# A CRISPR-based assay for the study of eukaryotic DNA repair onboard the International Space Station

**Sarah Stahl-Rommel[1], David Li[2]◉, Michelle Sung[3]◉, Rebecca Li[3]◉, Aarthi Vijayakumar[3]◉, Kutay Deniz Atabay[4,5]◉, G. Guy Bushkin[4,5]◉, Christian L. Castro[1], Kevin D. Foley[6], D. Scott Copeland[6], Sarah L. Castro-Wallace[7], Ezequiel Alvarez Saavedra[8], Emily J. Gleason[8]*, Sebastian Kraves[8]***

**1** JES Tech, Houston, Texas, United States of America, **2** Woodbury High School, Woodbury, Minnesota, United States of America, **3** Mounds View High School, Arden Hills, Minnesota, United States of America, **4** Massachusetts Institute of Technology, Cambridge, Massachusetts, United States of America, **5** Whitehead Institute for Biomedical Research, Cambridge, Massachusetts, United States of America, **6** Boeing Defense, Space & Security, Berkley, Michigan, United States of America, **7** Biomedical Research and Environmental Sciences Division, NASA Johnson Space Center, Houston, Texas, United States of America, **8** MiniPCR Bio, Cambridge, Massachusetts, United States of America

◉ These authors contributed equally to this work.
* emily@minipcr.com (EJG); seb@minipcr.com (SK)

**Data Availability Statement:** DNA sequence data will be available in the European Nucleotide Archive (ENA) under project PRJEB39039.

## Abstract

As we explore beyond Earth, astronauts may be at risk for harmful DNA damage caused by ionizing radiation. Double-strand breaks are a type of DNA damage that can be repaired by two major cellular pathways: non-homologous end joining, during which insertions or deletions may be added at the break site, and homologous recombination, in which the DNA sequence often remains unchanged. Previous work suggests that space conditions may impact the choice of DNA repair pathway, potentially compounding the risks of increased radiation exposure during space travel. However, our understanding of this problem has been limited by technical and safety concerns, which have prevented integral study of the DNA repair process in space. The CRISPR/Cas9 gene editing system offers a model for the safe and targeted generation of double-strand breaks in eukaryotes. Here we describe a CRISPR-based assay for DNA break induction and assessment of double-strand break repair pathway choice entirely in space. As necessary steps in this process, we describe the first successful genetic transformation and CRISPR/Cas9 genome editing in space. These milestones represent a significant expansion of the molecular biology toolkit onboard the International Space Station.

## Introduction

Astronauts traveling beyond the protective boundaries of Earth's magnetosphere are at an increased risk of DNA damage caused by ionizing radiation. Such DNA damage may lead to

**Funding:** This study was funded by miniPCR bio and Boeing.

**Competing interests:** E.A.S., E.J.G., and S.K. are employed by miniPCR bio, manufacturer of the device used for transformation, DNA extraction, and DNA amplification. D.S.C and K.D.F. are employed by Boeing and S.S.R. and C.L.C. are employed by JES Tech. This does not alter our adherence to PLOS ONE policies on sharing data and materials. The remaining authors declare no competing financial interests.

cancer and other detrimental health effects, raising questions about the safety of long-duration space travel [1].

Double-strand breaks (DSBs), in which the phosphate backbones of both DNA strands are hydrolyzed, are a particularly harmful type of DNA lesion [2]. On Earth, eukaryotic organisms use at least two mechanisms for repairing DSBs: homologous recombination (HR) and non-homologous end joining (NHEJ) [2]. During HR, a homologous DNA sequence is used as a template for repair so that the DNA sequence remains unchanged. HR repair is typically limited to the S and G2 phases of the cell cycle. NHEJ, however, can occur at any point in the cell cycle [3]. During NHEJ the cell directly rejoins the two pieces of DNA, often resulting in changes to the original DNA sequence [2, 4]. These alterations may increase the risk of cancer and other detrimental conditions.

In space, a significant portion of the ionizing radiation is Galactic Cosmic Radiation which is mainly composed of high linear energy transfer (LET) particles. These particles can create clustered and complex DNA damage that may be difficult to repair [5, 6]. Therefore, repair pathway choice may be especially important in mitigating damage from space radiation. For example, work by Zafar et al. suggests that HR is critical in repairing DNA damage caused by high-LET particles [7]. Previous studies have reported that the choice of DNA repair mechanism can be influenced by microgravity conditions. However, because of safety considerations and technological limitations, these studies have often relied on the generation of DSBs on Earth, after which the biological material was frozen and sent to space to assess DNA repair choice under microgravity conditions [1]. The initial recognition of the DNA break and the assembly of DNA repair factors at the break site are thought to be important determinants of repair pathway choice and may occur soon after the DSB [8, 9]. Therefore, it is possible that in previous studies of DNA repair in space the choice of repair occurred on Earth rather than in space. Furthermore, the rigors of space launch and associated handling introduce extraneous factors between induction of a DSB on Earth and eventual assessment of DNA repair [10]. We therefore sought to develop a method to study DSB break induction and repair entirely in the microgravity environment onboard the International Space Station (ISS) National Laboratory.

Our method relies on using a CRISPR-based mutagenesis strategy for the targeted generation of DSBs at a defined genomic locus. During CRISPR/Cas9 mediated genome editing, the Cas9 nuclease is directed by an engineered guide RNA to recognize and create a DSB at a specific site in the genome [2]. DNA repair mechanisms then make changes to the DNA sequence at the site of the DSB. NHEJ may introduce random insertions or deletions at the break site, while HR can be harnessed to make specific changes to the DNA sequence through an engineered repair template [11].

Using the CRISPR/Cas9 system to study DNA repair in space has several advantages over previously established models. First, this system does not utilize radiation or other reagents that cause widespread, non-specific DNA damage and are unsafe to use during spaceflight. Second, because the DSB is generated at a precise location in the genome, any changes in DNA sequence following repair can be easily identified and tracked using methods previously validated on the ISS, namely, polymerase chain reaction (PCR) and DNA sequencing [12–14].

Here we describe the first transformation of *Saccharomyces cerevisiae* with exogenous genetic material followed by the first CRISPR/Cas9 genome editing in space. These experiments were part of a complete workflow developed to allow the study of DNA repair entirely onboard the ISS National Lab. This workflow spans the targeted generation of DNA lesions in organisms living under microgravity conditions to the site-specific confirmation of DNA repair by molecular methods. Implementation of these techniques represents a significant expansion of the molecular biology toolkit onboard the ISS and lays the groundwork for future

experiments to address a sweeping array of questions and practical needs pertaining to space exploration and colonization.

## Results

### Adaptation of Earth methods for the study of DNA repair in space

We used a CRISPR mutagenesis strategy developed for use in *S. cerevisiae* [15]. In this system, auxotrophic *S. cerevisiae* lacking a functional *URA3* gene necessary for the biosynthesis of uracil are transformed with the pVG1 plasmid (Fig 1A). This plasmid expresses a functional *URA3* for positive selection of transformants, the Cas9 enzyme, and a guide RNA targeting Cas9 to the Adenine Requiring 2 (*ADE2*) gene. *S. cerevisiae* with *ADE2* mutations turn red because of the buildup of purine precursors in the vacuole, allowing for visual identification of mutant colonies [16]. The pVG1 plasmid additionally contains a repair template that introduces early stop codons into the *ADE2* gene (Fig 1A).

In pVG1-transformed *S. cerevisiae*, Cas9, directed by the guide RNA, introduces a DSB in the *ADE2* sequence leading to the induction of DNA repair. As *S. cerevisiae* relies heavily on homology directed repair, we hypothesized that transformed cells would utilize the repair template on the pVG1 plasmid to repair the DSB instead of NHEJ. Use of this template would result in a change at nucleotide 55 of the *ADE2* sequence where six base pairs of the wild type sequence are replaced with a 12 base pair sequence that includes two stop codons and an EcoRI restriction site [15]. Given that NHEJ repair would also introduce mutations at the break site, the *ADE2* gene will be mutated following CRISPR editing regardless of repair mechanism, resulting in a red phenotype and enabling visual identification of edited cells (Fig 1B).

The choice of repair pathway can be determined by analyzing the DNA sequence at the break site to determine whether the sequence includes the expected repair template sequence, indicating repair by HR, or whether the sequence contains random insertions or deletions, indicating NHEJ was used. This type of DNA sequence analysis can be performed using methods that have been previously established to work in space, specifically amplification by polymerase chain reaction (PCR) followed by nanopore sequencing [12–14, 17].

### Transformation of *Saccharomyces cerevisiae* onboard the International Space Station

The preparation of *S. cerevisiae* competent cells for this experiment was adapted from standard protocols to facilitate transport to and transformation onboard the ISS. Yeast BY4741 cells were grown on Earth to a concentration of approximately $1x10^8$ cells per milliliter, pelleted by centrifugation, and frozen at -80˚C for shipment to the ISS or kept as ground controls (Fig 1C). Onboard the ISS, and on the ground in parallel, pellets were thawed and resuspended in 100 mM lithium acetate, before being combined with a transformation mixture that included polyethylene glycol, lithium acetate, and the pVG1 plasmid (Fig 1A and 1C). The miniPCR thermal cycler [12] was then used as a heat block to induce transformation (Fig 1C).

Following transformation, cells were transferred to Petri dishes containing synthetic defined agar lacking uracil (SDA-URA plates) for growth and selection of transformants (Fig 1C). To plate the cells in microgravity, small volumes of liquid culture (~20 μl) were transferred to the surface of the agar using a micropipette and then spread using a sterile plastic spreader (Fig 2A, S1 Video). By transferring small volumes, the surface tension was sufficient to keep the yeast suspension attached to the agar before being spread by the force of a sterile plastic spreader. Ground controls were prepared in parallel using the same methods.

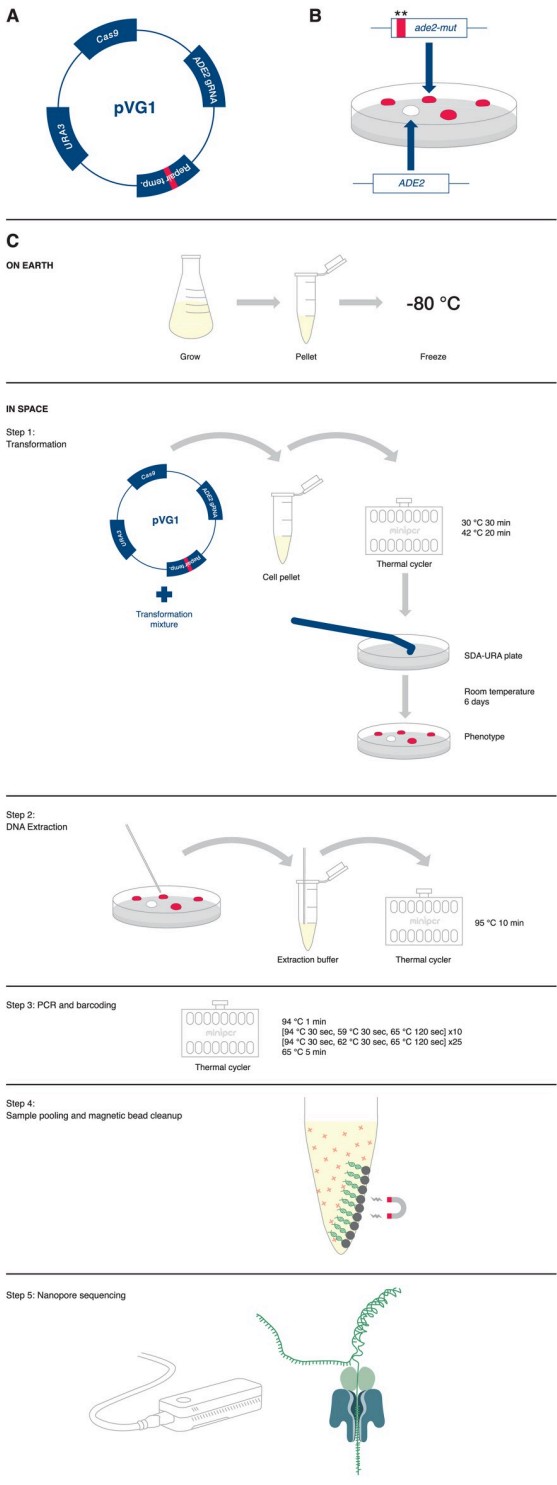

**Fig 1. Overview of CRISPR genome editing system adapted for use onboard the ISS.** A. Map of the pVG1 vector [15]. This vector contains CRISPR machinery: Cas9, guide RNAs targeting *ADE2*, a repair template that introduces two stop codons and an EcoRI site into the *ADE2* gene, and the *URA3* gene for positive selection. B. *ADE2* mutant colonies are easily distinguished from those bearing the wild type *ADE2* sequence. *ADE2* is not essential for survival, but *S. cerevisiae* with mutations in this gene turn red due to the buildup of purine precursors in the vacuole [16]. Wild type *S. cerevisiae* colonies are white. C. Adaptation of *S. cerevisiae* transformation and CRISPR/Cas9 genome editing protocols for use onboard the ISS. Prior to launch, cells were grown in liquid culture on Earth, pelleted by centrifugation, and frozen in glycerol at -80˚C for transport to the ISS. Step 1, transformation: transformation mixture

and pVG1 vector were added to thawed cells. The miniPCR thermal cycler was used as a heat block to induce transformation. Following transformation, cells were plated on synthetic defined agar lacking uracil (SDA-URA) and grown at room temperature for six days when the phenotype of the colonies was assessed. Step 2, DNA extraction: A pipette tip was used to transfer a small number of cells from four red and four white colonies to the DNA extraction buffer. Cells were heated in the miniPCR thermal cycler to 95°C to extract the DNA. Step 3, PCR and barcoding: DNA extract was directly added to PCR reagents. PCR was performed to amplify the 5' end of the *ADE2* gene. Sequencing barcodes were added at this step. Step 4, sample pooling and magnetic bead clean up: PCR product was pooled and purified during a magnetic bead cleanup step. Step 5, nanopore sequencing: Purified PCR product was sequenced by nanopore sequencing. Data was downlinked to the ground where sequences were assessed.

## Phenotypic assessment of CRISPR/Cas9 genome editing

The plates were assessed on the ISS and ground six days after transformation. The ISS crew reported four red colonies and six white colonies. Ground controls contained eight red and twenty-nine white colonies (Fig 2B–2E). The red phenotype is indicative of successful CRISPR/Cas9 mutagenesis resulting in the disruption of the *ADE2* locus (Fig 2D).

## Genotypic confirmation of CRISPR/Cas9 genome editing

To confirm successful CRISPR/Cas9 genome editing, the *ADE2* locus was examined using PCR and DNA sequencing. Four red colonies (labeled R1-R4) and four white colonies (labeled W1-W4) were randomly selected from both ISS and ground transformation plates for DNA extraction (Fig 2C and 2E). DNA was extracted from these colonies using a simple protocol where cells were heated to 95°C for 10 minutes in DNA extraction buffer.

Following DNA extraction, the 5' end of the *ADE2* sequence was amplified by PCR. PCR primers contained short barcode sequences; a total of four barcodes were used to identify the individual colonies sampled. Following amplification, the samples were combined into two pools of four samples each and purified using magnetic beads [17] (Fig 1C). Purified DNA was sequenced using the MinION nanopore sequencer (Fig 1C). Sequencing data was downlinked to Earth for analysis.

Sequencing of four pooled white colonies grown during spaceflight (flight) or on the ground yielded 1.3 million and 1.0 million total reads respectively, while sequencing of four pooled red colonies grown in flight or on the ground yielded over 5.0 million and 2.0 million total reads, respectively. Total read count varied between individual barcodes, however, all yielded sufficient quantities for analysis (S1 Table). Consensus sequences from seven of the eight white colonies aligned to the wild type *ADE2* sequence while consensus sequences from all eight of the red colonies aligned to the *ade2* repair template for both flight and ground (Fig 2F). The median alignment identity was 92% (S1 Table). However, two ground white colonies, W1 and W3, had unusually low coverage at some base pairs near the Cas9 cut site (Fig 2F). Sanger sequencing upon return to ground confirmed that ground W1 and W3 contained 9 base pair and 3 base pair deletions, respectively (S1 Fig).

The consensus sequence obtained from flight white colony W2 aligned to the *ade2* repair template sequence rather than to the wild type *ADE2* sequence (Fig 2F). However, Sanger sequencing results of this colony showed alignment to the wild type *ADE2* sequence as expected for colonies with a white phenotype (S1 Fig).

The four ground red colonies that were sequenced had between 88% and 100% coverage of the 12-nucleotide *ade2* repair reference template, while the four red colonies from flight had only 59–75% sequencing coverage (S2 Fig). Comparatively, the four white colonies from flight had 99–100% sequencing coverage of the *ADE2* wild type sequence, except W2 which aligned to the repair template and was more heterogeneous (S2 Fig).

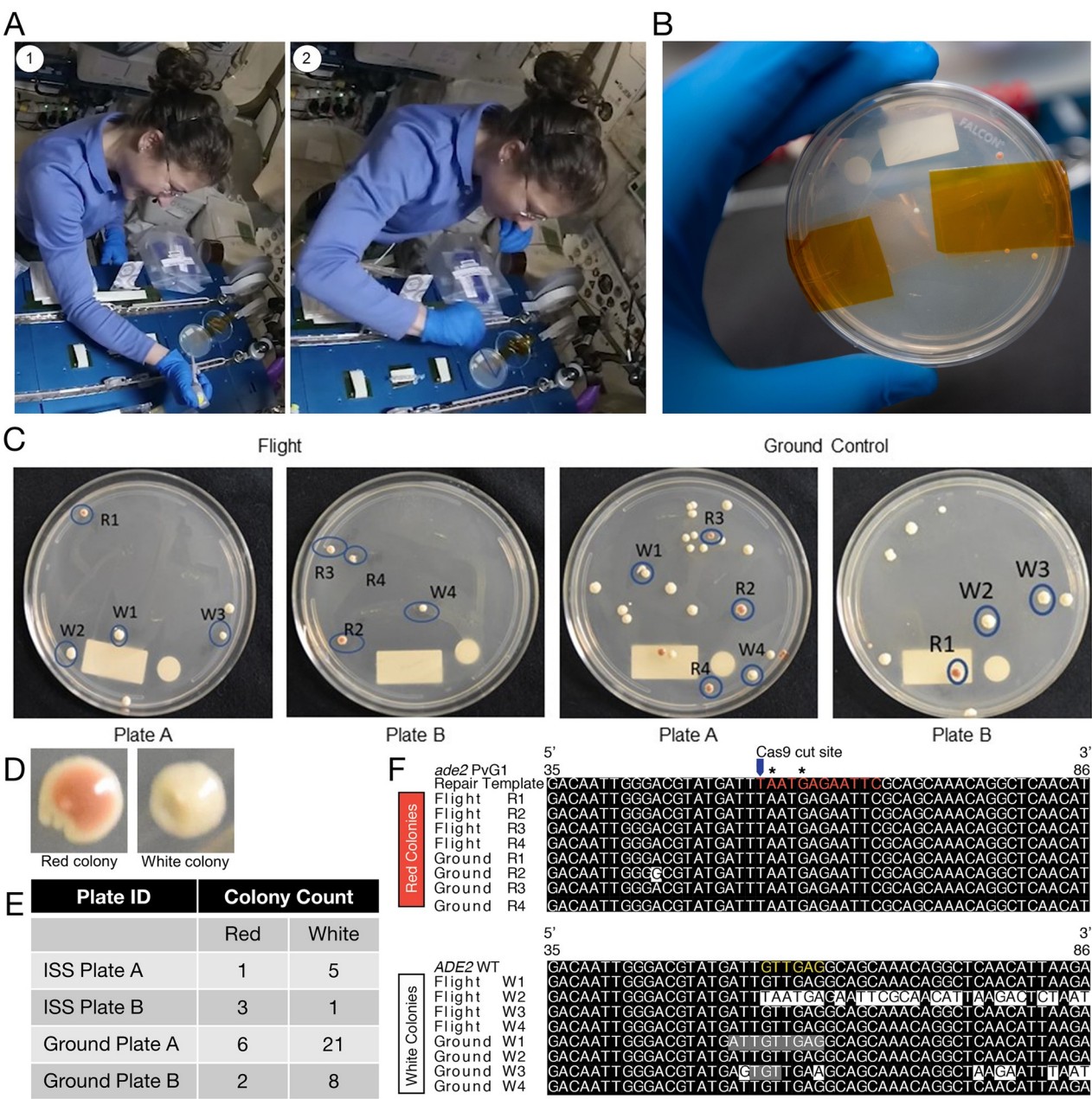

**Fig 2. Successful transformation and CRISPR/Cas9 genome editing onboard the ISS.** A. Astronaut Christina Koch plating *S. cerevisiae* following transformation (image credit: NASA). 1, Astronaut Koch transfers small volumes of liquid culture (~20 µl) onto the Petri dish multiple times so that liquid remains attached to the agar due to surface tension. 2, Cells were spread using a sterile plastic spreader. B. Astronaut Nick Hague examines a Petri dish following six days incubation at room temperature (image credit: NASA). Both white and red colonies are visible, suggesting successful CRISPR editing of the *ADE2* locus. C. Transformed *S. cerevisiae* colonies from flight and ground. Four red colonies, labeled R1-R4, and four white colonies, labeled W1-W4, were selected for further assessment by PCR and DNA sequencing. D. Examples of red and white colony phenotypes. Zoomed in photos of colony R2 and colony W4 from ground control plate A highlight the phenotypic differences between red and white colonies that make it easy to visually identify successfully CRISPR edited colonies. E. Total number of colonies of transformed *S. cerevisiae* seen in ground and flight experiments after six days of growth at room temperature. F. Alignment of nanopore sequences from red and white colonies transformed, cultured, extracted, and sequenced in flight or on the ground. Sequences are aligned to either the wild type *ADE2* sequence or the *ade2* repair template sequence. Red letters indicate the 12-base-pair insertion in the *ade2* repair template sequence. Stop codons are annotated with asterisks and the Cas9 cut site is indicated with an arrow. Yellow letters indicate the six base pairs found in the wild type sequence that are absent from the repair template. Black shading indicates bases that align to the reference sequence while white shading indicates a mismatch to the reference sequence. Gray shading indicates a match that has <50% coverage relative to other nucleotides.

## Discussion

Here we report the successful development of a complete molecular biology workflow for the assessment of DSB repair entirely onboard the ISS. The sequencing data presented here confirm the first transformation of live cells and the first CRISPR/Cas9 genome editing event in space (Fig 2F). In addition to enabling studies of DNA repair in microgravity, the ability to transform and genetically engineer organisms in space represents a significant advance and could enable a plethora of future investigations.

The challenges of adapting Earth-based protocols to the space environment are significant [17]. Microgravity poses challenges to liquid handling and safety concerns prohibit the use of equipment and reagents common in most laboratory settings on Earth. Therefore, the protocol described here required alterations to standard methods. For example, our transformation protocol reduced total sample volumes and used reagents that were premixed and frozen rather than prepared fresh as is usually done. This likely resulted in lower transformation efficiencies for both the flight and ground controls compared to what is typically seen in traditional *S. cerevisiae* transformation experiments. However, the transformation efficiency was still sufficient to allow the detection of CRISPR edited cells. Future studies could focus on improving the efficiency of the transformation protocol described here to allow a more extensive investigation of the frequency of HR compared to NHEJ DNA repair in microgravity conditions.

Although our protocol deviated from standard methods for the study of DNA repair on Earth, we find that it was generally sufficient to determine the mechanism of repair. In general, we obtained a lower number of reads from each red colony sequenced than was obtained from white colonies (S1 Table). This is likely due to the smaller colony size (Fig 2C). Regardless, our nanopore sequencing data is of sufficient quality to deduce the repair mechanism used by the yeast cells (Fig 2F, S1 Table). In this study, we confirmed that all red colonies sampled aligned to the repair template sequence, indicating that they were repaired using homologous recombination. These methods could enable quantification of repair pathway usage in future studies [15, 18].

Although our workflow represents a significant step towards enabling a better understanding of DNA repair pathway choice in microgravity conditions, this approach has one notable limitation. DSBs generated by Cas9 are much simpler than those generated by high-LET particles found in space [5]. As the complexity of the DNA damage may influence repair pathway choice, our model may not fully recapitulate the conditions found outside of Earth's magnetosphere [6]. Future studies could attempt to better mimic the effects of high-LET radiation by generating more complex breaks. For example, one might imagine creating clustered DNA damage sites using multiple gRNAs simultaneously.

Some noteworthy differences between flight and ground controls were observed. Initial sequencing data collected onboard the ISS for Flight W2, a colony with a white phenotype, aligned to the edited ADE2 repair template sequence. Subsequent Sanger sequencing analysis conducted upon return to ground confirmed the wild type genotype instead, as expected for a colony of the white phenotype (S1 Fig). It is likely that this discrepancy arose from a sampling or processing error during in-flight procedures. Additionally, nanopore sequencing data indicated that two ground control white colonies, ground W1, and ground W3, had lower coverage at several base pairs downstream of the Cas9 cut site, a finding suggestive of in-frame deletions (Fig 2F). Sanger sequencing confirmed the presence of these deletions (S1 Fig), ruling out sequencing or PCR errors. As the deletions are in-frame they do not appear to have altered the colony phenotype and may have been caused by Cas9 nuclease activity followed by NHEJ repair or by another form of mutagenesis. No such deletions were observed in our flight

samples. While the finding that indels occurred only in ground conditions might represent a finding of biological significance, the number of colonies sampled is too low to make any conclusions about the effects of space conditions on DNA repair pathway choice based on the current study alone.

Finally, the nanopore sequencing data from the flight and ground red colonies shows one intriguing difference. While greater than 90 percent of the reads from ground controls align to the *ade2* repair template sequence, the flight reads show more heterogeneity, with a significant portion aligning to the *ADE2* wildtype sequence (S2 Fig). This increased heterogeneity may simply be due to procedural differences in how the ISS crewmembers sampled the colonies compared to how they were sampled on the ground. Alternatively, this heterogeneity may reflect biological differences between the flight and ground CRISPR-edited colonies. Further study is needed to fully explain this observation.

Taken together, our results demonstrate the first successful use of both transformation and CRISPR/Cas9 genome editing in space and represent a significant expansion of the molecular biology toolkit onboard the ISS. In addition to establishing a viable platform for furthering our understanding of DNA repair in microgravity, these tools may enable the adaptation of many powerful methods for use in space. For example, the applications of CRISPR/Cas9 genome editing on Earth are rapidly expanding to include a number of gene editing approaches as well as novel uses of this technology such as viral detection [19]. One might imagine how CRISPR screens can expand our understanding of biological responses to microgravity or the utility of a simple detection assay in ensuring astronaut safety on long-duration missions. Similarly, genetic transformation of microbes has many applications, including the production of large amounts of a desired protein on demand. In the future, transformation in space might allow for on-demand production of critical medicines during deep space missions, or be used for pharmaceutical microgravity research in orbiting laboratories such as those currently in development by several commercial entities [20]. Our study has the potential to impact both our understanding of basic biological processes in microgravity as well as future space exploration and colonization and highlights the importance of basic molecular biology research onboard the ISS National Lab.

## Methods

### Preparation of *Saccharomyces cerevisiae* prior to spaceflight

*S. cerevisiae* strain BY4741 were grown overnight in yeast extract peptone dextrose (YPD) media (Thermo Fisher Scientific, Waltham, MA) at 30°C shaking at 110 rpm. The next morning, cells were diluted 1:30 in 50 ml of prewarmed YPD and grown to an O.D. 600 of ~1. Cells were pelleted for 3 minutes at 1,811 rcf, washed once with 1 ml 100 mM lithium acetate (MilliporeSigma, St. Louis, MO) with 10% glycerol, and transferred to a 1.5 ml microcentrifuge tube. Cells were collected at the bottom of the tube by spinning 30 seconds at 15,294 rcf. All liquid was removed and the cell pellet was placed into the -80°C freezer. The final concentration of the cells was approximately $1x10^8$ cells/ml.

### Preparation and handling of materials during spaceflight

The majority of the materials used in these experiments constituted Genes in Space-6 payload, which was launched to the International Space Station on NASA Commercial Resupply Services (CRS) mission 17 on May 4, 2019. Ambient temperature payload contained two mini16 miniPCR® thermal cyclers (miniPCR bio, Cambridge, MA), three magnetic separation stands (V&P Scientific, Inc., San Diego, CA), additional sterile individually packaged pipette tips (Eppendorf, Hamburg, Germany), and plate spreaders (Thermo Fisher Scientific). The

MinION sequencer (Oxford Nanopore Technologies (ONT), Oxford, England) and Research Plus Pipettes (Eppendorf) were already on board the ISS and available for payload use [14, 17]. All reagents were aliquoted into one-time use kits in flight certified tubes and stored at -80˚C for transport before being transferred into -80˚C Minus Eighty Degree Laboratory Freezer for ISS (MELFI). Petri dishes containing pre-poured agar were stored at 4˚C during transport and onboard the ISS. Additional ground control kits were prepared at the same time.

## Transformation and cell growth on the International Space Station

ISS crew member Christina Koch removed the pelleted *S. cerevisiae* from the MELFI, thawed, and resuspended it in 100 μl of 100 mM lithium acetate. 20 μl of cell suspension were added into PCR tubes containing 100 μl of ground prepared transformation mixture: 50% W/V poly-ethylene glycol (MilliporeSigma), 100 mM lithium acetate, 10 mM Tris pH 7.6 (Thermo Fisher Scientific), 1 mM EDTA (Thermo Fisher Scientific), 30 μg salmon sperm DNA (Thermo Fisher Scientific), and 500 ng of pVG1 plasmid (Addgene plasmid #111444, [15]). Reactions were tapped to mix and then heated using mini16 miniPCR® thermal cycler at 30˚C for 30 minutes and then 42˚C for 20 minutes. Upon cooldown, the crew member used a pipette to dispense 120 μl of transformed cells across a synthetic defined agar minus uracil (SDA-URA) plate (MP Biomedicals, Irvine, CA) and spread using a plastic spreader (Thermo Fisher Scientific) (S1 Video). Petri dishes were secured closed with tape, placed in re-sealable plastic bags, and stored at ambient temperature (approximately 20–25 ˚C) for six days. Ground controls were prepared following the same procedure. All conditions were prepared in duplicate.

## DNA extraction and PCR amplification

After a six-day incubation, NASA Astronaut Nick Hague observed each transformation plate, reported the total number of red or white colonies present, and photographed each plate. Four red (*ADE2* mutant) and four white (wild type *ADE2*) colonies were selected for DNA extraction and downstream sequencing analysis. Using a sterile pipette tip, the crew member lightly touched the colony and transferred cells into 100 μl X-Tract DNA extraction buffer (miniPCR bio). Cells were heated in the mini16 miniPCR® thermal cycler for 10 minutes at 95˚C. 10 μl of extracted DNA was loaded directly into PCR master mix containing LongAmp Taq 2X Master Mix (New England Biolabs (NEB), Ipswitch, MA), nuclease-free water, barcoded primers 1–8 from PCR Barcoding Kit PBK004 (ONT) and primers targeted to the 5' region of the *ADE2* gene spanning the region targeted by CRISPR (S2 Table). All red colonies were amplified using barcodes 1–4 while white colonies were barcodes 5–8. Barcoding followed the four primer PCR protocol (PBK004 Kit) as directed by Oxford Nanopore Technologies. For a more detailed explanation of Oxford Nanopore Technologies' barcoding strategies, see Matsuo et al., 2021 [21]. Amplification conditions were as follows: 94˚C 1 minute, [94˚C 30 seconds, 59˚C 30 seconds, 65˚C 60 seconds] x 10, [94˚C 30 seconds, 62˚C 30 seconds, 65˚C 60 seconds] x 25, 65˚C 5 minutes. PCR was run in duplicate on the ISS and one strip returned to ground as a backup. Ground observations, extraction, and PCR were completed in parallel.

## PCR clean up and DNA sequencing

The R9.4.1 MIN106 flow cell was prepared for sequencing with platform QC and bubble removal as in Burton et al. 2020 [14]. Flush buffer was prepared and the flow cell was washed twice to remove the storage buffer.

Following PCR amplification, the 8 PCR products were divided into two pools of 4 each for analysis following spaceflight sequencing protocols [17]. MinKNOW (version 2.2.15) was used

for sequencing for 24 hours on the Space Station Computer. Ground controls were treated identically in parallel.

## Sanger sequencing validation experiment

Upon return to ground, a single colony was used to inoculate SDA-URA liquid media. Cells were grown for 16 hours. An aliquot of 1.5 ml of growth culture was centrifuged at 750 rcf for 10 min to collect the cell pellet and DNA was extracted following the manufacturer's protocol (ZymoBIOMICS DNA Miniprep Kit, Zymo Research, U.S.A.). DNA was amplified using primers 306F and 308R (S2 Table) and Hot Start Long Amp Taq (NEB) as follows: 94°C for 1 minute, 94°C 20 seconds, 58°C 20 seconds, 65°C 20 seconds for 29 cycles with 5-minute extension at 65°C. The PCR amplicons were confirmed using 1% agarose gel electrophoresis, cleaned up using ExoSAP-IT Express (Thermo Fisher Scientific) following manufacturer's protocol, and quantified using Qubit 1X dsDNA HS assay kit (Thermo Fisher Scientific). DNA was aliquoted to approximately 10 ng per tube and sent to GENEWIZ (Cambridge, MA) for Sanger sequencing. Multiple sequence alignments were made with the consensus sequences using ClustalW (version 2.1). Sanger sequencing data is available in S3 Table.

## Nanopore sequencing data analysis

The fast5 files were basecalled using Oxford Nanopore Technologies Guppy command line tool (version 3.1.5). To assess the quality of the sequence data produced, FastQC (version 0.11.8, http://www.bioinformatics.bbsrc.ac.uk/projects/fastqc) and BasicQC (ONT) were used. The fastq reads were demultiplexed using qcat (version 1.0.1, https://github.com/nanoporetech/qcat) with the trim option enabled, and kit specific parameters. Reads were then subject to quality filtering using NanoFilt (version 2.5.0, https://github.com/wdecoster/nanofilt) to remove sequences less than 300 base pairs and greater than 1000 base pairs, and filtlong (version 0.2.0, https://github.com/rrwick/Filtlong) to remove the lowest 10% quality of reads. To generate consensus sequences, minimap2 (version 2.17) was first used to map the sequences to either the repair template or wild type *ADE2* sequence [22]. Bam files were then processed using bcftools (version 1.9) mpile up and call to generate vcf files [23]. The vcf files were then supplied to bcftools consensus to generate the final sequences. Multiple sequence alignments were made as described above. Visualizations of a region spanning 52 nucleotides for the *ADE2* repair template consensus sequences and 46 nucleotides for the *ADE2* wild type sequences were generated with Boxshade (version 3.31). Alignment identity was determined using marginStats (version 0.1, [24]). Coverage at each position was calculated using samtools depth [23]. DNA sequence data is available in the European Nucleotide Archive (ENA) under project PRJEB39039 (https://www.ebi.ac.uk/ena/browser/view/PRJEB39039).

## Supporting information

**S1 Video. Plating cells in microgravity.** Christina Koch first uses a pipette to dispense 120 μl of transformed *Saccharomyces cerevisiae* cells across the plate and then spreads them across the plate using a plastic spreader.
(PDF)

**S1 Fig. Sanger sequencing confirmation of nanopore sequencing results.** Colonies of interest were sequenced using Sanger methods and aligned to the expected *ADE2* wild type (WT) sequence. Flight W2 was returned to the ground from the ISS and the colony re-streaked on a fresh plate prior to sequencing. Sanger sequencing data from this colony aligns to the wild type sequence as expected from the white colony phenotype. Sanger sequences

show that Ground W1 contains a 9 bp deletion at position 52 and Ground W3 contains a 3 bp deletion at position 54. This supports nanopore sequencing data which showed low coverage at these positions.
(PDF)

**S2 Fig. Sequencing coverage plots of red and white colonies from flight and ground controls.** Nanopore reads are aligned to a hybrid reference sequence that contains both the *ADE2* wild type sequence (white bars) and repair template sequence (red bars) at 43–80 bp. The bar graph depicts the number of reads at each base pair, thus lower coverage corresponds with a lower number of total reads per sample. A. Red colonies sequenced in flight show heterogeneity. 63.2%, 58.8%, 72.6%, and 75.1% of the reads from flight samples R1- R4 respectively map to the sequence of the repair template, while 49.3%, 55%, 35.4%, and 31.1% of reads mapped to the wild type sequence. In contrast, data from red colony ground controls R1-R4 show that 97.9%, 90.8%, 96%, and 94% of the reads map to the repair template sequence and 10.4%, 16.2%, 11.7%, and 13% reads mapped to wild type sequence. B. With the exception of Flight W2 and Ground W1 and W3, >98% of reads map to the wild type sequence from both flight and ground. Flight W2 observed similar mapping as red colonies with 76% reads mapping to the repair template sequence and 33% of reads mapping to the wild type sequence. Ground W1 and W3 had lower coverage mapped to wild type sequence due to the observed deletions (S1 Fig).
(PDF)

**S1 Table. Nanopore sequencing metrics.** Includes total read count, demultiplexed read distribution, read length, and median alignment identity. A total of four nanopore sequencing runs were completed: pooled red colonies from flight, pooled white colonies from flight, pooled red colonies from ground, and pooled white colonies from ground. Reads were basecalled using Guppy and demultiplexed using Qcat. Read length determined with Samtools and Median Alignment Identity calculated from minimap2 alignments and Marginstats of red colonies to *ade2* repair template reference and white colonies to *ADE2* wild type reference.
(PDF)

**S2 Table. Primers used in this experiment.** AMP277 and AMP278 primers flank the Cas9 cut site specified by the *ADE2* guide RNA and contain adaptors for the LWB barcodes used in this study. AMP306 and AMP308 were used for Sanger sequencing confirmation of select colonies upon return to ground.
(PDF)

**S3 Table. Sanger sequencing data.** Sanger sequencing was used to verify Nanopore sequencing results for Ground W1, Ground W3, and Flight W2.
(PDF)

## Acknowledgments

We thank Boeing, miniPCR bio, The ISS National Lab, New England Biolabs, Inc., and MƒA for their support of the Genes in Space competition that led to this study. We thank Melissa Boyer, Teresa Tan, and Mohammad Abbas for their efforts in preparing and handling our materials for spaceflight and astronauts Christina Koch, Tyler Nicklaus Hague, and David Saint-Jacques for their work on this study. Finally, we thank Dr. Nicole Nichols of New England Biolabs, Inc. for contributing reagents for this study and Dr. Hang Nguyen for contributing Sanger sequencing data.

## Author Contributions

**Conceptualization:** David Li, Michelle Sung, Rebecca Li, Aarthi Vijayakumar, Kutay Deniz Atabay, G. Guy Bushkin, Ezequiel Alvarez Saavedra, Emily J. Gleason, Sebastian Kraves.

**Data curation:** Sarah Stahl-Rommel, Emily J. Gleason.

**Formal analysis:** Sarah Stahl-Rommel, Christian L. Castro, Sebastian Kraves.

**Investigation:** Sarah Stahl-Rommel, Kevin D. Foley, D. Scott Copeland, Sarah L. Castro-Wallace, Sebastian Kraves.

**Methodology:** Sarah Stahl-Rommel, David Li, Michelle Sung, Rebecca Li, Aarthi Vijayakumar, Kutay Deniz Atabay, G. Guy Bushkin, Kevin D. Foley, D. Scott Copeland, Sarah L. Castro-Wallace, Ezequiel Alvarez Saavedra, Emily J. Gleason, Sebastian Kraves.

**Project administration:** Ezequiel Alvarez Saavedra, Emily J. Gleason, Sebastian Kraves.

**Supervision:** Sebastian Kraves.

**Visualization:** Sarah Stahl-Rommel, Christian L. Castro, Emily J. Gleason.

**Writing – original draft:** Sarah Stahl-Rommel, Christian L. Castro, Emily J. Gleason, Sebastian Kraves.

**Writing – review & editing:** Sarah Stahl-Rommel, David Li, Michelle Sung, Rebecca Li, Aarthi Vijayakumar, Kutay Deniz Atabay, G. Guy Bushkin, Christian L. Castro, Kevin D. Foley, D. Scott Copeland, Sarah L. Castro-Wallace, Ezequiel Alvarez Saavedra, Emily J. Gleason, Sebastian Kraves.

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
