## [Decision Letter · Decision Letter 0]

23 Apr 2021

PONE-D-21-10318

A CRISPR-based assay for the study of eukaryotic DNA repair onboard the International Space Station

PLOS ONE

Dear Dr. Kraves,

Thank you for submitting your manuscript to PLOS ONE. After careful consideration, we feel that it has merit but does not fully meet PLOS ONE’s publication criteria as it currently stands. Therefore, we invite you to submit a revised version of the manuscript that addresses the points raised during the review process.

We look forward to receiving your revised manuscript.

Kind regards,

Ruslan Kalendar, PhD

Academic Editor

PLOS ONE

Journal Requirements:

PLOS requires an ORCID iD for the corresponding author in Editorial Manager on papers submitted after December 6th, 2016. Please ensure that you have an ORCID iD and that it is validated in Editorial Manager. To do this, go to ‘Update my Information’ (in the upper left-hand corner of the main menu), and click on the Fetch/Validate link next to the ORCID field. This will take you to the ORCID site and allow you to create a new iD or authenticate a pre-existing iD in Editorial Manager. Please see the following video for instructions on linking an ORCID iD to your Editorial Manager account: https://www.youtube.com/watch?v=_xcclfuvtxQ

Thank you for stating the following in the Financial Disclosure section:

This study was funded by Boeing (http://www.boeing.com/) and miniPCR bio (https://www.minipcr.com/). E.A.S., E.J.G., and S.K. are employed by miniPCR bio. D.S.C. and K.D.F. are employed by Boeing.

We note that one or more of the authors are employed by a commercial company: JES Tech, miniPCR bio and Boeing

3a, Please provide an amended Funding Statement declaring this commercial affiliation, as well as a statement regarding the Role of Funders in your study. If the funding organization did not play a role in the study design, data collection and analysis, decision to publish, or preparation of the manuscript and only provided financial support in the form of authors' salaries and/or research materials, please review your statements relating to the author contributions, and ensure you have specifically and accurately indicated the role(s) that these authors had in your study. You can update author roles in the Author Contributions section of the online submission form.

3b. Please also provide an updated Competing Interests Statement declaring this commercial affiliation along with any other relevant declarations relating to employment, consultancy, patents, products in development, or marketed products, etc. 

We note that you have stated that you will provide repository information for your data at acceptance. Should your manuscript be accepted for publication, we will hold it until you provide the relevant accession numbers or DOIs necessary to access your data. If you wish to make changes to your Data Availability statement, please describe these changes in your cover letter and we will update your Data Availability statement to reflect the information you provide.

Please upload a copy of Supporting Information Figure Video S1 which you refer to in your text on page 19.

We note that Figure 2 includes an image of a  participant in the study.

Reviewers' comments:

Reviewer's Responses to Questions

**Comments to the Author**

1. Is the manuscript technically sound, and do the data support the conclusions?

Reviewer #1: Yes

Reviewer #2: Yes

2. Has the statistical analysis been performed appropriately and rigorously? 

Reviewer #1: N/A

Reviewer #2: Yes

3. Have the authors made all data underlying the findings in their manuscript fully available?

Reviewer #1: Yes

Reviewer #2: Yes

4. Is the manuscript presented in an intelligible fashion and written in standard English?

Reviewer #1: Yes

Reviewer #2: Yes

5. Review Comments to the Author

Reviewer #1: 

In the article “A CRISPR-based assay for the study of eukaryotic DNA repair onboard the International Space Station” the authors report on a study on the DNA double strand break repair pathway selection in yeast on board the international space station using the CRISPR Cas9 system. Homologous Recombination is discriminated from Non-homologous end joining by next generation sequencing of the DSB site following DSB generation by the CRISPR Cas9 nuclease in the presence of an alternative homologous repair template. The experiments are accompanied by similar experiments conducted on earth to access differences in micro-gravity environment.

Overall the study is sound and demonstrates the future possibilities of molecular biology experiments on board the ISS. Due to the limitations in replicates the scientific outcome is limited. Nevertheless, considering the experimental limitations during space flight, I support the publication of the manuscript and only have some minor points.

Minor points:

Supplementary File naming as well as the authors list is not consistent with the main title.

Introduction, line 38 onwards: The authors should also mention the special type / complexity of DSBs generated by cosmic irradiation present in space.

Introduction, line 55 onwards: The authors should also include cell cycle specific aspects, when they discuss pathway selection DSB repair, especially in humans.

Results, line 103: I would replace he term “ADE2 mutant lines” with “ADE2 mutant colonies”

Results Figure 2: Since the colonies are not clearly visible (at least in the images provided to me) I would suggest to included a cropped inset of one colony each.

Methods, line 297: The authors should add the wavelength at with the O.D. was measured.

Methods, line 301-302: “Final cell pellets..” In my understanding the number of 1x10^8 cells is per pellet or total number, but not cells/ml. Or is this number representing the cell concentration after resuspension/reconstitution? The authors should clarify this point.

Methods, line 329 onwards: The authors describe that both the ground control as the in flight petri dishes were kept at room temperature. RT is not defined and should be given. Was it identical on the ISS and on the ground? If not this could explain the differences in colony size.

Methods, line 341: The bar-coding process should be explained in more detail, may be also with a small sketch, to help the less experienced reader to understand it.

Figure 1 has some elements at the end of the page that are probably misplaced.

Supplementary information: it reads “…Archive (ENA) under project PRJEB39039 (add link)”. No link is available and I could not find the mentioned project in ENA.

Reviewer #2: 

The manuscript by Stahl-Rommel and colleagues report the first attempt to undertake a simple DNA repair assay in space. This is important since an understanding of differences in maintaining genome stability on space have far-reaching implications for technologies that can employed in space and human health in space.

Sensibly, the team opted to use budding yeast as this is the most robust and best-characterised eukaryotic system available, coupled to an assay which suitable for studying the repair of targeted DNA double-strand breaks. The work is clearly presented and the caveats in interpreting the limited data obtained openly discussed.

I have only very minor comments:

1. I would suggest growing the yeast cells to a density of less than 1 x 108 CFU/mL (perhaps 2 x 107) might give a better transformation efficiency in future studies.

2. In the Introduction (p2, line 51) the possibility that microgravity conditions might influence DNA repair pathway usage or efficiency, which is an interesting possibility. It is also worth pointing out to readers that the ionising radiation background in space might damage DNA breaks close to their termini and this could influence repair efficiency. This is especially relevant during repair by HR, where tracts of ssDNA are generated as intermediates that might be more susceptible to such collateral damage. Such 'collateral' damage could plausibly influence the efficiency and outcome of repair.

6. PLOS authors have the option to publish the peer review history of their article (what does this mean?). If published, this will include your full peer review and any attached files.

Reviewer #1: No

Reviewer #2: **Yes: **Peter J.McHugh

---

## [Author Response · Author response to Decision Letter 0]

31 May 2021

To whom it may concern, 

We wish to thank the editor and both reviewers for their thoughtful comments our manuscript entitled “A CRISPR-based assay for the study of eukaryotic DNA repair onboard the International Space Station.” Below, we have included our responses to each point that was raised. We hope that we have satisfactorily addressed these concerns and that our manuscript is now suitable for publication. 

Sincerely,

Emily Gleason, Ph.D. Sebastian Kraves, Ph.D.

Academic editor: 

We have updated the title page to meet the title, author, affiliations formatting guide in both the manuscript. We additionally made changes to the way we referred to figures and tables within the manuscript text to better match the style guide. Finally, we have re-saved the supporting information as separate documents for each figure rather than as 1 compiled document and removed the author information from the names of all of our files. De-compiling the supporting information required us add S3 table that includes the Sanger sequencing data. We hope these changes are satisfactory and make our manuscript fit for publication. 

2) PLOS requires an ORCID iD for the corresponding author in Editorial Manager on papers submitted after December 6th, 2016. Please ensure that you have an ORCID iD and that it is validated in Editorial Manager. 

Sebastian Kraves’ ORCID iD is 0000-0002-6153-8029 Emily Gleason’s ORCID iD is 0000-0001-6875-8609. Emily Gleason’s was provided during the initial submission of the manuscript. 

3) Thank you for stating the following in the Financial Disclosure section:

This study was funded by Boeing (http://www.boeing.com/) and miniPCR bio (https://www.minipcr.com/). E.A.S., E.J.G., and S.K. are employed by miniPCR bio. D.S.C. and K.D.F. are employed by Boeing.

We note that one or more of the authors are employed by a commercial company: JES Tech, miniPCR bio and Boeing

a) Please provide an amended Funding Statement declaring this commercial affiliation, as well as a statement regarding the Role of Funders in your study. 

We have amended the statement to read as follows: 

This study was funded by Boeing (http://www.boeing.com/) and miniPCR bio (https://www.minipcr.com/). E.A.S., E.J.G., and S.K. are employed by miniPCR bio. D.S.C. and K.D.F. are employed by Boeing. S.S.R and C.L.C are employed by JES Tech (https://jestechllc.com/), but JES Tech played no role in funding this study. The specific roles of these authors are articulated in the ‘author contributions’ section. Boeing provided support in the form of salaries for authors D.S.C and K.D.F., but did not have any additional role in the study design, data collection and analysis, decision to publish, or preparation of the manuscript. The specific roles of these authors are articulated in the ‘author contributions’ section. miniPCR bio provided support in the form of salaries for authors E.A.S., E.J.G., and S.K., but did not have any additional role in the study design, data collection and analysis, decision to publish, or preparation of the manuscript. The specific roles of these authors are articulated in the ‘author contributions’ section.

We have updated the Competing Interests Statement to read as follows: 

E.A.S., E.J.G., and S.K. are employed by miniPCR bio, manufacturer of the device used for transformation, DNA extraction, and DNA amplification. D.S.C and K.D.F. are employed by Boeing and S.S.R. and C.L.C. are employed by JES Tech. This does not alter our adherence to PLOS ONE policies on sharing data and materials. The remaining authors declare no competing financial interests.

4) We note that you have stated that you will provide repository information for your data at acceptance. Should your manuscript be accepted for publication, we will hold it until you provide the relevant accession numbers or DOIs necessary to access your data. If you wish to make changes to your Data Availability statement, please describe these changes in your cover letter and we will update your Data Availability statement to reflect the information you provide.

We had uploaded the data to ENA, but kept it private pending the acceptance of the manuscript. The data has now been made public. You can find it in the ENA database under project PRJEB39039 (https://www.ebi.ac.uk/ena/browser/view/PRJEB39039). 

5) We note that Figure 2 includes an image of a participant in the study. Please amend the methods section and ethics statement of the manuscript to explicitly state that the patient/participant has provided consent for publication: “The individual in this manuscript has given written informed consent (as outlined in PLOS consent form) to publish these case details”. If you are unable to obtain consent from the subject of the photograph, you will need to remove the figure and any other textual identifying information or case descriptions for this individual.

The individual pictured is not a study participant, but rather Astronaut Christina Koch, one of the three astronauts who performed this work on the International Space Station. The images were supplied to us by NASA and as such, are not subject to copyright in the United States. NASA media guidelines state that these images may be used without explicit permission (https://www.nasa.gov/multimedia/guidelines/index.html). We have amended the figure legend to credit NASA for the images of Astronaut Koch as well as the image in Fig 2 B. We hope that this change is sufficient for these images to be included in the publication. 

To address concerns raised by the reviewers we have added the following references:

3. Zhao L, Bao C, Shang Y, He X, Ma C, Lei X, et al. The Determinant of DNA Repair Pathway Choices in Ionising Radiation-Induced DNA Double-Strand Breaks. Biomed Res Int. 2020;2020:4834965. 

5. Chancellor JC, Scott GB, Sutton JP. Space Radiation: The Number One Risk to Astronaut Health beyond Low Earth Orbit. Life (Basel). 2014;4(3):491-510.

6. Schipler A, Iliakis G. DNA double-strand-break complexity levels and their possible contributions to the probability for error-prone processing and repair pathway choice. Nucleic Acids Res. 2013;41(16):7589-605.

7. Zafar F, Seidler SB, Kronenberg A, Schild D, Wiese C. Homologous recombination contributes to the repair of DNA double-strand breaks induced by high-energy iron ions. Radiat Res. 2010;173(1):27-39.

21. Matsuo Y, Komiya S, Yasumizu Y, Yasuoka Y, Mizushima K, Takagi T, et al. Full-length 16S rRNA gene amplicon analysis of human gut microbiota using MinION nanopore sequencing confers species-level resolution. BMC microbiology. 2021;21(1):35.

We have checked our reference list. It is complete, correct, and, to the best of our knowledge, does not contain any retracted papers. 

Reviewer #1:

Minor points:

• Supplementary File naming as well as the authors list is not consistent with the main title.

We removed the author and title from the supporting information document and saved each figure and table individually rather than as 1 compiled document. We hope these changes address reviewer #1’s concerns. 

• Introduction, line 38 onwards: The authors should also mention the special type / complexity of DSBs generated by cosmic irradiation present in space.

This is a great suggestion. We have added a few sentences to address this to the introduction (lines78-83) and included three additional references (Chancellor et al., 2014, Schipler et al., 2013, Zafar et al. 2010). We hope this appropriately addresses reviewer 1’s comments. 

• Introduction, line 55 onwards: The authors should also include cell cycle specific aspects, when they discuss pathway selection DSB repair, especially in humans.

We have added some information regarding cell cycle impacts for repair pathway selection to the introduction (lines 72-74) and an additional reference (Zhao et al., 2020). 

• Results, line 103: I would replace he term “ADE2 mutant lines” with “ADE2 mutant colonies”

We have made the suggested change to the manuscript. 

• Results Figure 2: Since the colonies are not clearly visible (at least in the images provided to me) I would suggest to included a cropped inset of one colony each.

As there are colonies of both phenotypes on each plate, we were concerned about misleading people if we implemented the suggestion to choose one colony to include as an inset for each plate. Instead, we added a new subpanel (Fig. 2D) that gives a zoomed in example of each phenotype to illustrate the differences between edited and unedited colonies. We hope this addresses Reviewer #1’s concerns about the visibility of the colonies. 

• Methods, line 297: The authors should add the wavelength at with the O.D. was measured.

We used O.D. 600, this information has been added to the manuscript. 

• Methods, line 301-302: “Final cell pellets..” In my understanding the number of 1x10^8 cells is per pellet or total number, but not cells/ml. Or is this number representing the cell concentration after resuspension/reconstitution? The authors should clarify this point.

We have attempted to clarify this in the manuscript. 

• Methods, line 329 onwards: The authors describe that both the ground control as the in flight petri dishes were kept at room temperature. RT is not defined and should be given. Was it identical on the ISS and on the ground? If not this could explain the differences in colony size.

Ambient temperature is not precisely measured in either location, but would be between 20 and 25 ºC. We have noted this in the manuscript. 

• Methods, line 341: The bar-coding process should be explained in more detail, may be also with a small sketch, to help the less experienced reader to understand it.

We feel that explaining the details of the bar-coding process is a bit out of scope for this publication. Instead, we have added a reference to a source that explains this process in detail (lines 449-442, Matsou et. al. 2021). We hope this addresses reviewer 1’s concern. 

• Figure 1 has some elements at the end of the page that are probably misplaced.

This error occurred when switching file types. It has been corrected. 

• Supplementary information: it reads “…Archive (ENA) under project PRJEB39039 (add link)”. No link is available and I could not find the mentioned project in ENA.

See above comment. We had uploaded the data to ENA, but kept it private pending the acceptance of the manuscript. The data has now been made public. You can find it in the ENA database under project PRJEB39039 (https://www.ebi.ac.uk/ena/browser/view/PRJEB39039). 

Reviewer #2: 

I have only very minor comments:

• I would suggest growing the yeast cells to a density of less than 1 x 108 CFU/mL (perhaps 2 x 107) might give a better transformation efficiency in future studies.

We appreciate the suggestion. If we are given the opportunity to fly a follow up study, we will keep this in mind. 

• In the Introduction (p2, line 51) the possibility that microgravity conditions might influence DNA repair pathway usage or efficiency, which is an interesting possibility. It is also worth pointing out to readers that the ionising radiation background in space might damage DNA breaks close to their termini and this could influence repair efficiency. This is especially relevant during repair by HR, where tracts of ssDNA are generated as intermediates that might be more susceptible to such collateral damage. Such 'collateral' damage could plausibly influence the efficiency and outcome of repair.

This is a great point and similar to one of reviewer 1’s comments. We added a few sentences to the introduction (lines 78-83) and three new references (Chancellor et al., 2014, Schipler et al., 2013, Zafar et al. 2010). We additionally added a paragraph to the discussion that addresses the differences between the DNA lesions created by Cas9 and those thought to be created by galactic cosmic radiation (lines 323-330). We hope this sufficiently addresses reviewer 2’s comment.

---

## [Editor Report · Decision Letter 1]

4 Jun 2021

A CRISPR-based assay for the study of eukaryotic DNA repair onboard the International Space Station

PONE-D-21-10318R1

Dear Dr. Kraves,

We’re pleased to inform you that your manuscript has been judged scientifically suitable for publication and will be formally accepted for publication once it meets all outstanding technical requirements.

Kind regards,

Ruslan Kalendar, PhD

Academic Editor

PLOS ONE

---

## [Editor Report · Acceptance letter]

9 Jun 2021

PONE-D-21-10318R1 

A CRISPR-based assay for the study of eukaryotic DNA repair onboard the International Space Station 

Dear Dr. Kraves:

I'm pleased to inform you that your manuscript has been deemed suitable for publication in PLOS ONE. Congratulations! Your manuscript is now with our production department. 

Kind regards, 

on behalf of

Prof. Ruslan Kalendar 

Academic Editor

PLOS ONE